Osmotrophic glucose and leucine assimilation and its impact on EPA and DHA content in algae

http://orcid.org/0000-0002-2858-8358 Peltomaa Elina T. 1 elina.peltomaa@helsinki.fi
Taipale Sami 2
1 Faculty of Biological and Environmental Sciences, Lammi Biological Station, University of Helsinki , Lammi , Finland
2 Department of Biological and Environmental Science, University of Jyväskylä , Jyväskylä , Finland
Berges John
Electronic publication date: 2020 Jan 3
Publication date: 2020
Volume: 8
Electronic Location ID: e8363
Received 2019 Sep 9; Accepted 2019 Dec 6
Copyright: © 2020 Peltomaa and Taipale
Copyright year: 2020
Copyright holder: Peltomaa and Taipale
License: This is an open access article distributed under the terms of the Creative Commons Attribution License, which permits unrestricted use, distribution, reproduction and adaptation in any medium and for any purpose provided that it is properly attributed. For attribution, the original author(s), title, publication source (PeerJ) and either DOI or URL of the article must be cited.
License URL: https://creativecommons.org/licenses/by/4.0/

Keywords: Mixotrophy, Omega-3 fatty acids, Stable isotope labeling, Cryptophytes

Funding: Academy of Finland 276268 This work was supported by the Academy of Finland (No. 276268). The funders had no role in study design, data collection and analysis, decision to publish, or preparation of the manuscript.

==============================
The uptake of dissolved organic compounds, that is, osmotrophy, has been shown to be an efficient nutritional strategy for algae. However, this mode of nutrition may affect the biochemical composition, for example, the fatty acid (FA) contents, of algal cells. This study focused on the osmotrophic assimilation of glucose and leucine by selected seven algal strains belonging to chlorophytes, chrysophytes, cryptophytes, dinoflagellates and euglenoids. Our laboratory experiments with stable isotope labeling showed that osmotrophy occurred in four of the selected seven strains. However, only three of these produced long chain omega-3 FAs eicosapentaenoic acid (EPA; 20:5ω3) and docosahexaenoic acid (DHA; 22:6ω3). High glucose content (5 mg L−1) affected negatively on the total FAs of Mallomonas kalinae and the total omega-3 FAs of Cryptomonas sp. Further, glucose assimilation explained 35% (negative effect) and leucine assimilation 48% (positive effect) of the variation of EPA, DHA and the FAs related to their synthesis in Cryptomonas sp. Moderate glucose concentration (2 mg L−1) was found to enhance the growth of Cryptomonas ozolinii, whereas low leucine (20 µg L−1) enhanced the growth of M. kalinae. However, no systematic effect of osmotrophy on growth rates was detected. Our study shows that osmotrophic assimilation of algae is species and compound specific, and that the effects of the assimilated compounds on algal metabolism also varies depending on the species.

Introduction

Mixotrophy, that is, the ability of an organism to combine autotrophy and heterotrophy and thus get sustenance simultaneously from inorganic and organic sources, is gaining increasing attention in studies of aquatic as well as terrestrial ecosystems. In aquatic habitats, mixotrophy is common in unicellular organisms such as algae, cyanobacteria and protists (Flynn et al., 2013; Schmidt, Raven & Paungfoo-Lonhienne, 2013). The heterotrophic nutrition in mixotrophy is via phagotrophy (particle or cell uptake) or osmotrophy (uptake of dissolved organic compounds), both of which may occur in tandem with photosynthesis or during dark periods, for example, nights.

The detection of mixotrophic behavior can be challenging both in laboratory and field conditions, and thus, for example, radioactive isotope labeling has been applied to study the osmotrophic nutrient uptake (Kamjunke & Tittel, 2008; Tittel et al., 2009; Beamud et al., 2014). These studies have shown that many algae have the ability to assimilate carbohydrates (e.g., glucose), amino acids (e.g., glutamine, leucine, thymidine, aspartic acid) and other organic compounds (e.g., acetic acid, coumaric acid, glycerol), which they use as carbon and nitrogen sources, and which are commonly released by the algae themselves or by bacteria (Hellebust, 1965; Kamjunke & Tittel, 2008; Tittel et al., 2009; Beamud et al., 2014; Dąbrowska, Nawrocki & Szeląg-Wasielewska, 2014). Osmotrophy has been shown to be an efficient nutrition strategy for algae in nature: osmotrophic assimilation of amino acids prevent nitrogen limitation, which favors biomass growth in oligotrophic lakes (Kamjunke & Tittel, 2008). Similarly, osmotrophic uptake of fulvic acids enhances biomass growth and boost bloom forming in humic lakes (Rengefors et al., 2008).

Osmotrophic nutrition may affect the biochemical composition of algal cells. Biosynthesis of various molecules is determined by phylogeny-based traits and there is a significantly different composition of, for example, fatty acids (FAs) in different algal taxa (Kohli et al., 2016). It is reported that growth conditions account for relatively low variation in algal FAs compared to phylogeny (Galloway & Winder, 2015); however, studies with Ochromonas sp. (Boëchat et al., 2007) showed decreased polyunsaturated fatty acid (PUFA) concentration by feeding mode. Thus, even though the FA profiles and the quality of synthetized FAs may not change, the quantity of different FAs might be affected by the growth mode. For example, nitrogen limitation favors FA synthesis and lipid accumulation in algal cells, and thus if algae can assimilate leucine and use it as their nitrogen source, they should not start to accumulate lipids but carry on cell division as long as the other essential nutrients are available. In turn, osmotrophically assimilated glucose is channeled directly into lipid synthesis, that is, to palmitic acid (16:0), which results in building up of triacylglycerols and for example, long chain polyunsaturated FAs (LC-PUFAs; Ratledge, 2004). Some LC-PUFAs belong to omega-3 FAs (e.g., eicosapentaenoic acid (EPA; 20:5ω3) and docosahexaenoic acid (DHA; 22:6ω3)), and for their part are of utmost importance for the growth and reproduction of consumers in aquatic food webs (Peltomaa et al., 2017; Taipale et al., 2018). Since algae are practically the only primary source of EPA and DHA in aquatic food webs (Colombo et al., 2017), osmotrophic nutrient uptake may affect the whole food web by influencing the availability of these nutritionally essential compounds, and thus the growth and reproduction of upper trophic levels (Jonasdottir, 1994; Brett et al., 2006; Peltomaa et al., 2017; Taipale et al., 2018).

In this study we focused on the osmotrophic uptake of glucose and leucine by selected seven algal strains belonging to chlorophytes, chrysophytes, cryptophytes, dinoflagellates and euglenoids. We conducted short-term stabile isotope labeling experiments with these algae to determine if they are able to assimilate glucose and/or leucine. Since we were especially interested in the effects of osmotrophy on EPA and DHA production, we analyzed the FAs of these strains. For studying the impact of osmotrophy on FA synthesis, we selected the EPA and DHA synthesizing osmotrophic strains and cultured them with glucose, leucine and mix of these two in a long-term experiment for 14 days, that is, until the cultures reached the stationary or late exponential phase, during which the LC-PUFAs are mobilized from the membranes into storage lipids (Roessler, 1990; Boelen et al., 2017). We hypothesized that (1) all of the strains are osmotrophic, that is, assimilate glucose and leucine, (2) osmotrophy has positive effect on their growth, and that (3) the osmotrophic uptake of glucose increases the EPA and DHA concentrations in algae capable of synthesizing these FAs, whereas (4) the uptake of leucine does not affect specifically their EPA and DHA concentrations, but may actually lower their total FA content.

Materials and Methods

The algal strains and growth conditions

The studied algal strains were from freshwater origin, and included chlorophytes Chlamydomonas reinhardtii (from the collection of the University of Washington, UWCC, Seattle, WA, USA) and Selenastum sp. (SCCAP K-1877), chrysophyte Mallomonas kalinae (SCCAP K-1759), cryptophytes Cryptomonas sp. (CPCC 336) and C. ozolinii (UTEX LB 2782), dinoflagellate Peridinium sp. (author’s collection, isolated from Lake Valkea-Kotinen, Finland, 61.14°N, 25.04°E in 2015) and euglenoid Euglena gracilis (CCAP 1224/5Z). The stock cultures of the strains were grown autotrophically in AF6 medium (E. gracilis; Watanabe et al., 2000) or MWC medium (all the other strains; Guillard & Lorenzen, 1972) at 20 °C under light:dark cycle of 16:8 h with light intensity of 70–100 μmol m−2 s−1. The cultures were not axenic, but the initial numbers of bacteria were low due to the growth media consisting of inorganic nutrients. The bacterial numbers were not determined, but bacterial FA biomarkers were included in the FA analysis (see below). The algal cultures were grown to late exponential phase before they were used in the experiments.

Short-term stable isotope labeling experiments

For the short-term stable isotope experiment, the autotrophically grown seven algal strains were grown to stationary phase, collected with centrifugation into pellets (200 mL, 5 min, 2,000 rpm, which was pre-examined as safe for the fragile flagellates), and further resuspended into 50 mL of fresh AF6 (E. gracilis) or MWC media (all the other strains) enriched with glucose 5 mg L−1 (containing 4.9 mg L−1 non-labeled D-(+)-Glucose (Sigma–Aldrich Co., St. Louis, MO, USA) and 0.1 mg L−1 13C-labeled D-Glucose (Sigma–Aldrich Co., St. Louis, MO, USA)) or with leucine 400 µg L−1 (containing 392 µg non-labeled L-Leucine (Sigma–Aldrich Co., St. Louis, MO, USA) and 8 µg 15N-labeled L-Leucine (Sigma–Aldrich Co., St. Louis, MO, USA)) or with both. Three independent replicates were used for each algae and treatment, and non-labeled autotrophic controls were run in parallel. The incubation took place at 20 °C under a constant light intensity of 70–100 μmol m−2 s−1, that is, the possible dark-time heterotrophic assimilation of glucose and leucine was excluded from this short-term experiment. The incubation time was only 30 min to preventing respiration loss of the labels, but it was still long enough to acquiring detectable changes in cellular δ13C and δ15N concentrations. After the incubation, the samples were centrifuged (2,000 rpm, 5 min), the supernatants were discarded, and the pellets were flushed by diluting them into 30 mL of fresh AF6 (E. gracilis) or MWC media (all the other strains) and centrifuging them again. After discarding the supernatants, the pellets were frozen in −80 °C, and freeze-dried within 2 days. The δ13C and δ15N as well as FA profiles were analyzed from these samples.

The δ13C and δ15N analyses

For the δ13C and δ15N analyses, approximately 2 mg of the freeze-dried algal biomass was weighed into tin capsules. The analyses were carried out on a Carlo-Erba Flash 1112 series Element Analyzer connected to a Thermo Finnigan Delta Plus Advantage IRMS (Thermo Fisher Scientific, Waltham, MA, USA). Four replicates were run from each sample. The samples were compared to the NBS-22 standard using birch leaf powder as a laboratory-working standard. The precision of the δ13C and the δ15N analyses were 0.2% and 0.3%, respectively, for all samples.

Fatty acid analysis

Two replicates of each freeze-dried sample were weighed (1–2 mg/sample) into tin capsules and the lipids were extracted using chloroform:methanol (2:1) NaCl-method (Parrish, 1999). Toluene and sulfuric acid were used for the transesterification of fatty acid methyl esters (FAMEs) at 90 °C for 1 h. The FAMEs were analyzed with a gas chromatograph (Shimadzu Ultra, Kyoto, Japan) equipped with a mass detector (GC-MS; Shimadzu Ultra, Kyoto, Japan) and using helium as a carrier gas and an Agilent® (Santa Clara, CA, USA) DB-23 column (30 m × 0.25 mm × 0.15 µm). FA concentrations were calculated using calibration curves based on known standard solutions of a FAME standard mixture (GLC standard mixture 566c, Nu-ChekPrep, Elysian, MN, USA) (see Taipale et al., 2016 for further details). The 16:0, alpha-linolenic acid (ALA; 18:3ω3), stearidonic acid (SDA; 18:4ω-3), EPA and DHA concentrations, and the total sum of monounsaturated FAs with 16 carbons (16 Monounsaturated fatty acids (MUFAs)) and 18 carbons (18 MUFAs) were in the focus of this study and thus reported here. The FA biomarkers for bacteria (i-14:0, i-15:0, a-15:0, i-16:0, i-17:0 and a-17:0; Brennan, 1989; Taipale et al., 2015) were also detected in order to ensure that the numbers of bacteria were low, that is, the glucose and leucine were assimilated by the algae, not by bacteria.

The long-term osmotrophy experiments

The long-term experiments were done only for those three strains that were detected to be osmotrophic (i.e., assimilated either glucose or leucine or both in the short-term experiment) and, based on the results from the short-term study, were detected to synthetize either EPA or DHA or both. These strains were: M. kalinae, Cryptomonas sp. and C. ozolinii. The long-term experiments were run in independent triplicates, and using three different glucose (0.5, 2 and 5 mg L−1) and leucine (20, 100 and 400 µg L−1) concentrations selected based on literature (Kamjunke et al., 2008, Kamjunke & Tittel, 2008). There were no mixed treatments of glucose and leucine, but autotrophic controls were run in parallel. The algal cells were collected into pellets from the stock cultures similarly to short-term experiments before transferring them into the experimental flasks of 250 mL. The strains were grown for 15–16 days in similar conditions as the algal stock cultures and the growth was followed through microscopic counts every 3rd day using Sedgewick Rafter–counting cells and preservation with acid Lugol’s solution (Willén, 1962). The specific growth rates (µ; d−1) for all strains were calculated using the Eq. (1). The cells were grown into the stationary or late exponential phase (Fig. S1), harvested during the light-period of the light:dark-cycle, and pelleted, frozen (−80 °C) and freeze-dried. The FA profiles were analyzed similarly to the short-term samples (see above), but only from two replicates of each treatment and from one control.

(1) μ=ln(cells Tx/cells T0)/(Tx−T0)

where:

μ is the specific growth rate

cells T0 is the cell number at time 0 (T0)

cells Tx is the cell number at time x (Tx)

Statistical analyses

The results of the short-term isotope labeling experiments were statistically tested with t-test by comparing the non-labeled autotrophic control samples with the labeled samples. The effects of osmotrophy on growth and FA contents were in the long-term experiment tested with the analysis of variance (ANOVA) and Tukey’s honestly significant difference (HSD) post hoc test. Levene’s test was used for testing the homogeneity of variances. Principal component analysis (PCA), permutational multivariate analysis of variances (PERMANOVA) and similarity percentages (SIMPER) were used for a more detailed study of the similarity of the FA profiles between the treatments in the long-term experiment. In PCA the 16:0, ALA, 18:3ω3; SDA, 18:4ω-3; EPA and DHA concentrations, and the total sum of monounsaturated FAs with 16 carbons (16 MUFAs) were included in the analysis. In PERMANOVA and SIMPER the analysis included the above mentioned FAs and also MUFAs with 18 carbons (18 MUFAs). All glucose treatments and all leucine treatments were pooled for the analysis in PERMANOVA and SIMPER. PERMANOVA was run with unrestricted permutation of raw data and type III sums of squares. Monte Carlo adjustment was used in PERMANOVA due to low numbers of replicates. In the statistical testing, p-values < 0.05 were considered as significant. ANOVA and Tukey’s and Levene’s tests were conducted with IBM SPSS Statistics for Windows, Version 22.0 (IBM Corp., Armonk, NY, USA). PCA, PERMANOVA and SIMPER were done using Primer 7 (version 7.0.13; Quest Research Limited, Auckland, New Zealand).

Results

The osmotrophic glucose and leucine uptake

The studied strains showed different responses to glucose and leucine additions in the short-term experiment (Fig. 1; Table S1). The chlorophyte C. reinhardtii, the dinoflagellate Peridinium sp. and the euglenoid E. gracilis did not show glucose or leucine assimilation at all. Whereas, the chlorophyte Selenastrum sp. assimilated both compounds (glucose t-test p < 0.001, leucine t-test p = 0.04), as did also the cryptophytes Cryptomonas sp. (glucose t-test p < 0.01, leucine t-test p = 0.02) and C. ozolinii (glucose t-test p < 0.01, leucine t-test p < 0.001). The chrysophyte M. kalinae did not assimilate glucose, but assimilated leucine (t-test p < 0.01).

Figure 1 Osmotrophic glucose and leucine assimilation.

The osmotrophic glucose and leucine assimilation in the studied algal strains was detected with stable isotope labeling. The figures show the isotopic difference in 13C (A) and 15N (B) between the treatments in Cryptomonas sp., Cryptomonas ozolinii, Chlamydomonas reinhardtii, Euglena gracilis, Mallomonas kalinae, Peridinium sp. and Selenastrum sp. in the short-term experiment. The cultures were inoculated with 13C-labeled glucose, 15N-labeled leucine and mixture of these two. The total concentration of glucose in the experiment was 5 mg L−1 and leucine 400 mg L−1. Replication n = 3. The bars show standard errors, statistically significant difference between the non-labeled and labeled treatments are marked with star symbols (ANOVA, * p < 0.5, ** p < 0.1 and *** p < 0.01).

Fatty acid profiling of the short-term experiment

The FA profiling of the short-term experiments showed that the strains capable of EPA and/or DHA synthesis were M. kalinae (average EPA 0.3% of all FAs, sd 0.0; average DHA 3.2% of all FAs, sd 0.1), Cryptomonas sp. (EPA 17.6%, sd 0.0; DHA 2.3%, sd 0.0), C. ozolinii (EPA 13.9%, sd 1.3; DHA 2.1%, sd 0.2), Peridinium sp. (EPA 13.4%, sd 0.0; DHA 25.1%, sd 0.6) and E. gracilis (EPA 14.3%, sd 0.8; DHA 8.4%, sd 0.2). However, because only M. kalinae, Cryptomonas sp. and C. ozolinii were showing osmotrophic uptake of glucose and/or leucine, these three strains were selected for the long-term experiment. FA biomarkers for bacteria (i-14:0, i-15:0, a-15:0, i-16:0, i-17:0 and a-17:0) were not found in the analysis.

The effect of osmotrophy on growth, total FAs and EPA and DHA production

In general, C. ozolinii had higher specific growth rates than the two other studied strains in the long-term experiment. The highest growth rate in C. ozolinii (µ = 0.88 d−1; ANOVA p < 0.01) was detected with moderate glucose concentration of 2 mg L−1, and in M. kalinae (µ = 0.47 d−1, ANOVA p < 0.01) with low leucine concentration of 20 µg L−1, but otherwise there were no signs that the osmotrophic nutrition would have had increased or decreased the growth rates of the studied three strains (Table 1).

Table 1 Growth rates.

The maximal specific growth rates (µ; d−1) of Mallomonas kalinae, Cryptomonas sp. and C. ozolinii in the long-term experiment. The growth rates were calculated for the exponential growth phase using Eq. (1). Replication n = 3.

Strain	Control	Glucose 0.5	Glucose 2	Glucose 5	Leucine 20	Leucine 100	Leucine 400	
Mallomonas kalinae	0.20 (0.03)	0.20 (0.08)	0.23 (0.01)	0.20 (0.01)	0.47 (0.05)*	0.20 (0.04)	0.16 (0.06)	
Cryptomonas sp.	0.21 (0.04)	0.23 (0.06)	0.36 (0.05)	0.26 (0.02)	0.23 (0.04)	0.25 (0.05)	0.28 (0.03)	
Cryptomonas ozolinii	0.52 (0.01)	0.41 (0.01)	0.88 (0.07)*	0.34 (0.07)	0.34 (0.05)	0.65 (0.05)	0.57 (0.02)	
Note:

* ANOVA p < 0.01, standard deviations are given in parenthesis.

The highest total FA concentration (i.e., 88.8 µg FA in mg DW) was found in the autotrophic M. kalinae, and the total FAs of M. kalinae were also detected to vary according to the treatment that was lowest (33.8 µg FA in mg DW) in the 5 mg L−1 glucose treatment (Table 2). However, it could not be firmly stated that either glucose or leucine addition would have had certain effects on the total FA content in M. kalinae. There were no treatment-based variation in the total FAs of the two cryptophytes either, but there was species specific variation: the total FA concentrations in Cryptomonas sp. (52.8–73.6 µg FA in mg DW) were substantially higher in all treatments than in C. ozolinii (range 32.9–45.1 µg FA in mg DW; Table 2). FA biomarkers for bacteria (i-14:0, i-15:0, a-15:0, i-16:0, i-17:0 and a-17:0) were not found in the FA analysis.

Table 2 Fatty acid results.

The concentrations (µg in mg dry weight) of 16:0 fatty acid (FA), alpha-linolenic acid (ALA; 18:3ω3), stearidonic acid (SDA; 18:4ω3), eicosapentaenoic acid (EPA; 20:5ω3), docosahexaenoic acid (DHA; 22:6ω3), total omega-3 (ω-3) FAs and total FAs in M. kalinae, Cryptomonas sp. and C. ozolinii in autotrophic controls and when grown osmotrophically with different glucose (i.e., 0.5 mg L−1, 2 mg L−1, 5 mg L−1) and leucine (20 mg L−1, 100 mg L−1, 400 mg L−1) concentrations in the long-term experiment. Replication n = 3. Standard deviations are given in parenthesis. Different letters (a, b, c) denote significant differences (Tukey’s HSD p < 0.05) between treatments, only statistically significant results are indicated.

Strain	Fatty acid	Control	Glucose 0.5	Glucose 2	Glucose 5	Leucine 20	Leucine 100	Leucine 400	
Mallomonas kalinae	16:0	8.2 (2.3)	4.1 (0.1)	7.9 (1.5)	5.0 (1.9)	8.1 (1.0)	4.0 (0.1)	6.8 (1.9)	
	ALA	7.3 (0.0)a	6.7 (4.1)ab	8.3 (0.0)b	4.8 (4.7)abc	10.9 (1.1)b	4.6 (0.3)c	8.0 (1.1)ab	
	SDA	10.3 (1.6)a	8.3 (6.3)a	9.0 (1.4)a	6.2 (5.4)ab	12.9 (1.2)a	5.6 (0.4)b	10.6 (3.9)ab	
	EPA	0.3 (0.0)a	0.2 (0.1)ab	0.3 (0.0)a	0.2 (0.1)ab	0.3 (0.0)a	0.1 (0.0)b	0.3 (0.0)a	
	DHA	0.8 (0.0)	0.7 (0.5)	0.8 (0.1)	0.6 (0.3)	1.0 (0.1)	0.4 (0.0)	0.7 (0.2)	
	Total FA	88.8 (5.8)a	70.3 (8.7)ab	74.1 (18.9)ab	33.8 (14.8)bc	63.8 (27.9)ab	45.6 (0.3)b	56.8 (22.3)ab	
Cryptomonas sp.	16:0	7.8 (1.0)ab	7.8 (1.6)ab	13.6 (0.4)b	8.6 (0.7)ab	10.4 (1.2)ab	7.6 (2.0)ab	10.2 (0.3)a	
	ALA	21.7 (3.7)ab	16.8 (2.3)ab	23.2 (0.3)a	18.9 (3.2)ab	20.1 (0.9)ab	17.1 (1.2)ab	18.0 (0.2)b	
	SDA	11.5 (2.2)	14.9 (1.3)	17.1 (2.6)	15.1 (3.8)	20.8 (2.2)	13.3 (4.0)	23.0 (1.2)	
	EPA	13.1 (2.1)a	11.5 (0.5)ab	13.6 (3.3)ab	12.3 (2.7)ab	14.9 (2.5)ab	11.1 (3.0)ab	16.8 (0.4)b	
	DHA	1.4 (0.0)	2.0 (0.2)	2.1 (0.6)	1.8 (0.6)	2.8 (0.3)	1.7 (0.6)	3.4 (0.2)	
	Total FA	60.7 (2.1)	52.8 (16.5)	73.6 (3.1)	70.5 (15.6)	67.7 (16.2)	55.9 (7.8)	68.8 (26.1)	
Cryptomonas ozolinii	16:0	4.3 (0.0)	5.2 (1.5)	4.8 (1.5)	4.6 (0.2)	5.3 (1.0)	4.5 (0.2)	5.0 (0.1)	
	ALA	8.1 (1.3)	10.1 (3.6)	9.5 (4.1)	9.0 (0.8)	10.9 (2.1)	9.2 (0.5)	8.5 (2.8)	
	SDA	8.5 (0.2)	11.7 (4.2)	10.9 (6.1)	11.1 (0.8)	13.6 (1.7)	11.5 (1.2)	10.0 (3.7)	
	EPA	7.0 (2.6)a	8.9 (2.8)ab	8.5 (4.5)ab	9.1 (1.8)ab	11.1 (0.1)b	9.7 (1.6)ab	8.2 (2.5)ab	
	DHA	1.6 (0.0)a	2.2 (0.7)ab	2.0 (1.2)ab	2.4 (0.7)ab	2.8 (0.0)b	2.6 (0.5)ab	2.0 (0.7)ab	
	Total FA	35.9 (7.1)	35.2 (0.3)	44.8 (18.0)	37.1 (4.6)	45.1 (16.9)	43.0 (9.6)	32.9 (2.8)	

The proportion of omega-3 FAs (i.e., ALA, SDA, EPA and DHA) out of total FAs varied between the strains being lowest in M. kalinae (range 21.3–40.3%), and was substantially higher in Cryptomonas sp. (range 76.8–89.9%) and in C. ozolinii (69.9–94.8%; Fig. 2). The PCA plot of the omega-3 FAs clustered the strains into their own groups in spite of the growth conditions (Fig. 3) indicating strong genetic control of FA profiles and synthesis. When EPA and DHA were studied more in detail, no clear evidence of the effects of osmotrophy on the contents of these FAs were found with ANOVA: in M. kalinae EPA and its precursor ALA varied between the treatments, in Cryptomonas sp. variations were found in 16:0, ALA and EPA, and in C. ozolinii in EPA and DHA, but either glucose or leucine could not be stated to have specific effect on these omega-3 FAs (Table 2; Table S2). However, when the 16:0, ALA, SDA, EPA, DHA, 16 MUFA and 18 MUFA concentrations were studied with PERMANOVA, the glucose assimilation were found to explain 35% (PERMANOVA, F(3, 6) = 2.74, p = 0.025) and the leucine assimilation 48% (PERMANOVA, F(3, 6) = 3.78, p = 0.021) of the variation in these FAs in Cryptomonas sp. (Table 3). Statistically significant results were not found with PERMANOVA for M. kalinae or C. ozolinii (Table 3). These observations are in line with the results of the PCA (Fig. 3), indicating that Cryptomonas sp. differ from M. kalinae, but also from C. ozolinii.

Figure 2 Fatty acid contents.

The proportions of omega-3 FAs (alpha-linolenic acid, ALA; 18:3ω3, stearidonic acid, SDA; 18:4ω3, eicosapentaenoic acid, EPA; 20:5ω3, docosahexaenoic acid, DHA; 22:6ω3) on total omega-3 FAs in (A) M. kalinae, (B) Cryptomonas sp. and (C) C. ozolinii in autotrophic controls and when grown osmotrophically with different glucose (i.e., 0.5 mg L−1, 2 mg L−1, 5 mg L−1) and leucine (20 mg L−1, 100 mg L−1, 400 mg L−1) concentrations in the long-term experiment. Replication n = 3.

Figure 3 PCA plot.

Principal component analysis (PCA) plot of the 16:0 fatty acid (FA), alpha-linolenic acid (ALA; 18:3ω3), stearidonic acid (SDA; 18:4ω3), eicosapentaenoic acid (EPA; 20:5ω3) and docosahexaenoic acid (DHA; 22:6ω3) of the long-term experiment showing that the studied three strains (Mallomonas kalinae, Cryptomonas sp. (CPCC 336) and C. ozolinii) differ from each other based on these FAs despite of the growth conditions (autotrophic, or osmotrophic with glucose or leucine; data shown in Table S2).

Table 3 PERMANOVA results.

Permutational multivariate analysis of variances (PERMANOVA) results for comparisons of the similarity of the concentrations of selected FAs (16:0, ALA, SDA, EPA, DHA, 16 MUFA, 18 MUFA) between the treatments in the long-term experiment. For the analysis, all glucose and all leucine treatments were pooled, thus replication n = 9. SS, sum of squares; MS, mean squares; P (perm) significance; P (MC) significance after Montecarlo correction. Statistically significant results are bolded.

Strain	Treatment	df	SS	MS	Pseudo-F	P (perm)	P (MC)	
Mallomonas kalinae	Glucose	3	1697.1	565.71	1.4768	0.253	0.271	
	Leucine	3	1564.9	521.63	1.3617	0.331	0.339	
Cryptomonas sp.	Glucose	3	375.57	125.19	2.7389	0.025	0.047	
	Leucine	3	518.04	172.68	3.7779	0.021	0.03	
Cryptomonas ozolinii	Glucose	3	117.55	39.184	0.20727	0.918	0.898	
	Leucine	3	339.31	113.1	0.59828	0.702	0.654	

Discussion

In this study, we focused on the osmotrophic nutrition and omega-3 FA production of seven algal strains representing chlorophytes, chrysophytes, cryptophytes, dinoflagellates and euglenoids. We expected that all of the studied strains would assimilate glucose and leucine (Hypothesis 1), but this was not the case. The chlorophyte C. reinhardtii, the dinoflagellate Peridinium sp. and the euglenoid E. gracilis did not assimilate either glucose or leucine during the 30 min incubation period of our short-time experiment. Furthermore, the chrysophyte M. kalinae assimilated only leucine. The uptake velocities of different algae may vary for different compounds (North & Stephens, 1972) and depending on the growth mode (Wheeler, North & Stephens, 1974), and thus it is possible that even though our cultures were on stationary phase and assumingly depleted by nutrients, our 30 min incubation time was not sufficient enough for assimilation for some of the algae used in this study. We did the incubations in light, which also could have affected the results, since in some cases algae have shown higher osmotrophic assimilation in dark than in light for glucose (Beamud et al., 2014) and leucine (Ruiz-González et al., 2012). However, many algal species are reported to enhance their osmotrophic uptake in light (Tittel et al., 2009; Beamud et al., 2014). Additionally, our experiments were done in nutrient-rich AF6 or MWC medium, but inorganic nutrient limitation could have triggered rapid osmotrophic uptake (Kamjunke & Tittel, 2008). However, it has also been shown by Beamud et al. (2014) that osmotrophic feeding mode is not triggered only by nutrient deficiency: in their study the chlorophytes Keratococcus rhaphidioides and Watanabea sp. assimilated leucine, thymidine, aspartic acid and acetate also under high levels of inorganic nitrogen and phosphorus, and not only during nutrient limitation. Altogether, our study and the previous observations show that osmotrophic assimilation of algae is both species and compound specific, and that generalizations on the occurrence of osmotrophy within certain taxa cannot be done.

For studying the effects on osmotrophy on algal FAs, especially the LC-PUFAs, the FA profiles of the strains were screened in the short-term study. It is already known that most EPA and DHA producing algae belong to the kingdom Chromista, that is, to cryptophytes, haptophytes and heterokonts (Cavalier-Smith, 2010; Mühlroth et al., 2013). This was the case also with our algae: EPA and DHA was found in all strains excluding the chlorophytes C. reinhardtii and Selenastrum sp. However, since we specifically focused on the effects of osmotrophic nutrition on EPA and DHA production, only M. kalinae and the two cryptophyte strains showing osmotrophic assimilation were studied in our long-term experiment. In this experiment, we expected that (Hypothesis 2) the growth rates would have been enhanced by the osmotrophic growth mode, as has earlier been reported for the raphidophyte Gonyostomum semen (Rengefors et al., 2008) and for the chlorophytes K. rhaphidioides and Watanabea sp. (Beamud et al., 2014). This however was not the case, and excluding some statistically significant differences in the growth rates of M. kalinae and C. ozolinii, the growth rates could not be directly related to glucose or leucine assimilation. We cannot fully explain the reason behind these two observations on higher growth rates, but are aware that for example, mixotrophic phagotrophy on bacteria may increase algal growth (Yoo et al., 2017). We did not observe high amounts of bacteria in the samples during the algal cell counts or detect bacterial FA biomarkers in the FA analysis. However, we did not calculate the bacterial numbers nor study the possible assimilation of bacteria by the algal strains, and thus cannot explicitly state that phagotrophic mixotrophy did not occur during the long-term experiment.

Because glucose is channeled directly to palmitic acid (16:0) and further into lipid synthesis (Ratledge, 2004), and the cellular neutral lipid content should be at highest during the light period of the light:dark cycle (Roessler, 1990), and because the LC-PUFA content should be at highest during the stationary growth phase (Roessler, 1990; Boelen et al., 2017), we expected (Hypothesis 3) that the osmotrophic uptake of glucose increases the FA content in algae. However, we did not find any specific effect of glucose on the amount of total FAs or EPA or DHA. Our PERMANOVA analysis for EPA and DHA, and the FAs related to the synthesis of these LC-PUFAs (16:0, ALA, SDA, 16 MUFA, 18 MUFA), showed that the glucose assimilation explained (35%) the concentrations of these FAs in Cryptomonas sp., but the effect was rather negative than positive (Fig. 2). It has been shown earlier that too high glucose concentration may inhibit growth and lipid synthesis and that the optimal glucose content is species specific (Liang, Sarkany & Cui, 2009; Wan et al., 2011). This effect was seen besides in the omega-3 in Cryptomonas sp. also in the total FA content in M. kalinae, which had lowest total FAs in the highest glucose treatment (5 mg L−1).

In contrast to glucose addition, we expected that (Hypothesis 4) the leucine addition would not affect the EPA and DHA content of algae, but could boost their growth and thus simultaneously actually reduce the amount of stored FAs. However, the growth rates were not affected by leucine, and reduction in the total FAs compared to control was found only in M. kalinae in leucine 100 µg L−1 treatment. For our surprise, in PERMANOVA, leucine assimilation explained 48% of the variation in the content of 16:0, ALA, SDA, EPA, DHA 16 MUFA and 18 MUFA of Cryptomonas sp., and—unlike expected—the effect of leucine was positive. Again, these results show that biochemical synthesis in algae is species specific, and that generalizations cannot be made.

We selected the glucose and leucine concentrations based on literature (Kamjunke & Tittel, 2008; Kamjunke et al., 2008), and they are in line with the dissolved organic carbon (DOC) contents of natural lakes; the DOC content in the clearwater lakes in Finland vary between 7 and 9 mg C L−1 (Ojala et al., 2011; Bręk-Laitinen, López Bellido & Ojala, 2012), whereas in humic lakes the DOC values can be even higher (10–45 mg C L−1; Taipale et al., 2008; Ojala et al., 2011). However, in nature the DOC consists of both recalcitrant compounds from mainly terrestrial origin and labile compounds released by algae and bacteria. Further, the labile compounds constitute of different carbohydrates, organic acids, dissolved and free amino acids, ketones and aldehydes with variable concentrations (Hellebust, 1965; Norrman et al., 1995; Peltomaa & Ojala, 2010; Dąbrowska, Nawrocki & Szeląg-Wasielewska, 2014), which makes the detection of osmotropic assimilation as well as the evaluation of its effects on for example, FA synthesis challenging. In this study, we found some positive and some negative effects of osmotrophic assimilation on FA synthesis, but the effects were still minor in general, which agrees with the study of Galloway & Winder (2015), who reported that growth conditions account for relatively low variation in algal FAs. However, in extreme conditions, for example, during enhanced run-off (either due to climate change or seasonality) leading to higher carbon or amino acids in the water, the magnitude of the effects on LC-PUFA availability could be significant at food web level (from algae to fish; Jonasdottir, 1994; Brett et al., 2006; Peltomaa et al., 2017; Taipale et al., 2018) assuming that the LC-PUFA producers of the algal community would consist of species capable on osmotrophic uptake of these compounds.

Conclusions

Our experiments show that osmotrophic nutrition can be found in different types of algae, but the assimilation is species specific and may differ between different organic compounds, as shown here with glucose and leucine. Furthermore, the effects of these two compounds on the algal growth and metabolism was found to be species specific: moderate glucose concentration (2 mg L−1) enhanced the growth of C. ozolinii, whereas the growth of M. kalinae was enhanced by low leucine (20 µg L−1). Additionally, high glucose content (5 mg L−1) affected negatively on the total FAs of M. kalinae and the total omega-3 FAs of Cryptomonas sp. In general, glucose assimilation explained 35% (negative effect) and leucine assimilation 48% (positive effect) of the variation of EPA, DHA and the FAs related to their synthesis in Cryptomonas sp. but not in the other algae studied. The broad spectrum of compounds and the species-specific responses of algae makes the estimation of the importance of osmotrophy challenging in planktonic food webs and natural waters in general.

Supplemental Information

Supplemental Information 1 Stable isotope labeling results.

The carbon and nitrogen drift and drift percentage values for the studied seven algal strains.

Click here for additional data file.

Supplemental Information 2 Fatty acid data for the long-term experiment.

The concentrations of the 16:0, ALA, SDA, EPA, DHA and total FAs in the controls and in different glucose (i.e., 0.5 mg L−1, 2 mg L−1, 5 mg L−1) and leucine (20 mg L−1,100 mg L−1,400 mg L−1) concentrations.

Click here for additional data file.

Supplemental Information 3 The growth curves of the long-term experiment.

Cryptomonas sp. when grown with different concentrations of glucose (A) and leucine (B), Cryptomonas ozolinii with glucose (C) and leucine (D), and Mallomonas kalinae with glucose (E) and leucine (F). Error bars show the standard errors.

Click here for additional data file.

The authors would like to thank the reviewers for all of their careful, constructive and insightful comments in relation to this work, Miss Paula Ilut for helping with the algal culturing and Mr. Roy Nyberg for editing the English of this manuscript.

Additional Information and Declarations

Competing Interests

Author Contributions

Data Availability

The authors declare that they have no competing interests.

Elina T. Peltomaa conceived and designed the experiments, performed the experiments, prepared figures and/or tables, authored or reviewed drafts of the paper, and approved the final draft.

Sami Taipale analyzed the data, prepared figures and/or tables, authored or reviewed drafts of the paper, and approved the final draft.

The following information was supplied regarding data availability:

The raw measurements are available in the Supplemental Files.

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
