# Peer review of "Osmotrophic glucose and leucine assimilation and its impact on EPA and DHA content in algae"

_PeerJ, doi:10.7717/peerj.8363_

## Round 0.1 · original submission · Major Revisions

The manuscript has now been assessed by two reviewers with somewhat different expertise. Both find merit in the work and recommend publication, but have identified some issues that need attention. On balance, I am considering these “major” revisions because they will require a degree of re-thinking and re-writing that goes beyond grammar and presentation. But, I anticipate that these should not require extra experimentation or extensive re-analysis.

Please carefully examine the Reviewers comments and reply to them on a point-by point basis, describing the changes you have made, or, in the case where you disagree with the suggestions, explaining your rationale and the clarifications you have made (in the case where the Reviewer has misunderstood a point).

Reviewer 1 has also provided an edited manuscript to assist you in identifying and improving some grammatical issues.

From my perspective, a few issues to high-light:

1) Both reviewers have, directly or indirectly, identified the significance of the phase of culture growth on the results. This needs to be addressed. The phase of culture growth at harvest needs to be better defined (see Reviewer 1 comments), perhaps giving the cell numbers at that point. From the growth rates quoted, a reader can easily calculate that cultures aren’t capable of staying in exponential phase for 15-16 days. Then, in the discussion, some attention needs to be devoted to the potential effects of growth phase on both osmotrophic ability and fatty acid composition. The Reviewers have provided some suggestions.

2) Reviewer 2 points out the potential significance of bacteria in the cultures and asks for clarification as to whether the cultures were axenic and how this was verified. If bacteria were present, they could also have taken up glucose and leucine and produced fatty acids, so the issue is especially important to address and discuss.

3) Reviewer 2 asks for clarification on culture replication. Cultures inoculated independently and grown in different flasks represent true replicates, while multiple samples from a single culture are, by definition, pseudo-replicates.

4) It isn’t specified at any point whether the algae are freshwater or marine, or a mixture. This would be useful.

5) The units of growth rate presented are incorrect. As presented in equation 1, µ (the coefficient of exponential growth) will have units d-1, NOT divisions per day. If a natural logarithm base is used (ln), then divisions per day can be calculated by dividing µ by ln(2). Thus µ = 0.88 d-1 is about 1.27 divisions per day. This is a critical point and needs to be corrected throughout the manuscript and Tables.

6) There are a number of confusing statistical results that appear to be inconsistent and require checking and perhaps discussion. A) For example, Results presented in lines 182-189 suggest that M. kalinae is assimilating 15N leucine, yet to judge by error bars in Figure 1 B, this isn’t so. In fact, I can’t see any statistically significant 15N assimilation noted in Figure 1 B. If this is, in fact, significant, then it certainly looks as through Selenastrum sp. is taking up 15N leucine, too. B) Differences that are detected and supported by Tukey tests need to cite that test…frequently the statistic cited is for an ANOVA, which is incapable of showing such individual differences. C) Later in the manuscript, the contrast between the PERMANOVA analysis and conventional ANOVA analysis seem to surprise the authors. It’s worth evaluating the power of the test and the faith one can put in the results. Some discussion should be devoted to this. A reader unfamiliar with PERMANOVA might for example want some insight into the “Monte Carlo” adjustment of P and whether choosing a P-value of 0.05 is wise for such a design. In contrast, I think Reviewer 1 is less familiar with presentation of results of multiple comparisons; the notation used in Table 2 appears fine to me, except that some rows have no notation. If there are no differences, shouldn’t all have an “a”? Otherwise, explain in the legend.

7) Standard deviations should be added to the growth rates in Table 1.

8) There are number of spelling and grammar issues that the Reviewers have picked up on. Overall, the grammar in the manuscript is reasonably good. One recurring issue is use of the adjective article “the”; it’s frequently inserted when not needed. e.g. line 35. not needed before “algal”; line 59. not needed at start of sentence; line 78. Not needed before “upper”, etc. Another issue is use of prepositions, e.g. “ to” instead of “for” in lines 117 and 118, “to” instead of “on” in line 183, “of” instead of “on” in line 220, etc. I appreciate that these are some of the most difficult (and inconsistent) elements of English. A read-through by a native speaker (even one unfamiliar with science) should catch them.

·

Basic reporting

• This is an interesting manuscript on how osmotrophic growth affects fatty acid composition of microalgae. This is of interest to scientific readers from various backgrounds, including ecology and biotechnology and as such, I recommend it for publication following amendments and clarification.
• The text is generally well written, however some improvement on the language and sentence structure will help to ensure the points are delivered in a more concise and structured manner.
• The introduction was very good; however, a mention towards the role of algal lipids in the food webs was introduced, however not re-visited in the discussion. I believe a paragraph in the discussion on what the possible implications of this research are on aquatic food webs will enhance your impact. For example, if there is enhanced run-off (either due to climate change, or seasonality) leading to higher carbon / amino acids in the water, this might lead to a reduction in total fatty acids of certain algae species (e.g. in this study) and therefore reduced transfer along the food web.
• The figures are relevant to the manuscript and illustrate the points made. For figure 1 I would like it if the authors highlighted what the isotopic difference is relative to (I am guessing the internal standards) to make this clear.
• In Table 2 I am unsure why the authors have used different letters to denote significant differences, and then do not explain what each means. For instance a, ab, b, c, bc, abc (as in table 1, I would stick with using an asterisk to denote significance, and you could use ** or *** to denote higher significance), but I would also add to the figure legend what the letters or number of asterisks means.

Experimental design

The research questions within this manuscript have been well defined, and I believe it fills an important knowledge gap.
Specific comments regarding the methods section are included below.
• Line 100; authors used Peridinium sp. from their own collection, but it would be useful to have information on where this organism was collected from, when etc. This allows comparison with other literature
• Line 108; the authors used centrifugation to collect the algae, do they know if this had any impact on algal physiology? With some strains, we sometimes find that centrifugation can lead to low recovery.
• Line 115; I would just specify here that for the short-term experiments it was; “under a constant light intensity of…”
• Line 154; it is specified that the algal strains were grown for 15-16 days, however no mention is made to what phase of the growth cycle they were in when the cells were harvested. If it is exponential or stationary this can vastly affect the fatty acid profiles (see for isntance Boelen et al., 2017, Growth phase significantly decreases the DHA-to-EPA ratio in marine microalgae, Aquaculture International, 25 (2)). It would be nice to see (even in supplementary) the growth curves for the algae used in the long-term osmotrophy experiment. This could also be one reason why we see such different profiles for M. kalinae as algae tend to accumulate FAs in stationary phase. Similarly, what time of the light: dark cycle were the algae harvested at, as this can also affect FA profiles.
• Line 158; The equation is given but not what the terms mean. I would just add a line saying “..using the equation 1, where Tx is the number of elapsed days during the experiment and cellsTx the number of cells, whereas T0 is at the start of the experiment etc..” just to ensure clarity to the reader.
• Line 160; it is mentioned see the method in chapter 2.4, and in this paper chapters aren’t used, so I would change this to say “see section above” or something similar.

Validity of the findings

The data provides an interesting look at how different algal strains react to changes in glucose and leucine concentrations. This is of interest both for ecologists, as it can affect fatty acids and thus the transfer of energy rich material through the food chain, and also biotechnologists who wish to optimise yield in industrial processes. This paper makes an interesting point that the response of algae is both species and compound-specific, and I think this is an important finding for researchers’ and industry. I have made some suggestions below on some extra discussion points and questions below.
• Line 192; it is mentioned that fatty acid profiling was conducted on all algae strains during the short-term experiment, however I cannot see this data is displayed anywhere (including the supplementary information).
• Were duplicate measurements of freeze-dried autotrophic controls conducted? If so I cannot see this data presented.
• Given some of the variability seen between duplicates, I would have recommended triplicate measurements of the samples (e.g. total FAs for Cryptomonas sp. at 0.5 glucose was 64.4 and 41.2 µg in mg DW). If it is possible to run triplicates based on preserved material this would be great.
• If any of the algae were in stationary phase when harvesting, then this should be mentioned in a discussion section as a reason for any differences seen in lipid profiles.
• Line 268; can the authors provide any hypothetical reasons for why some growth enhancement was seen during some experiments for M. kalinae and C. ozolinii? For instance possible enhanced bacterial growth at concentrations used leading to either enhanced mutualistic interactions, or possible phagotrophy of the bacteria themselves (which has been observed for cryptophytes – see Yoo et al., 2017. Mixotrophy in the marine red-tide cryptophyte Teleaulax amphioxeia and ingestion and grazing impact of cryptophytes on natural populations of bacteria in Korean coastal waters, Harmful Algae, 68, 105-117).
• Line 293; The first line of the conclusion says that osmotrophic nutrition was wide-spread and found in different types of algae, however in this study only 3 of the 7 strains tested displayed this mode of nutrition. This disagrees with line 234-235 of the discussion and hypothesis 1.
• The glucose concentrations used within this experiment seem very high for typical DOC concentrations, however I am not sure if this is true for freshwater systems. It would be useful if the authors in the discussion could try to contextualise the concentrations used to natural systems (e.g. is 5 mg L-1 what would be observed during highly eutrophic conditions with high runoff, or is it an extreme situation?).
• Are any of the algae strains used within this study of interest for biotechnological applications? For instance Cryptomonas sp, and can any recommendations for these processes be made (even if it is just to try other organic compounds or repeat with different concentrations etc..).
• It can be interesting to conduct some multivariate statistics which allow visualisation of the differences between algal strains and treatments. For instance, a principle component analysis using the lipid groups analysed within this study. I had a quick attempt at this using the data provided within Table 2 of the manuscript, and attach a file with the PCA plot output with my review as a pdf (I am happy to share the code if required also). I think this nicely shows that the Cryptomonas sp. is very different to the other 2 algae (although all are dissimilar in some way), but they also show the widest response with glucose and leucine treatment (as shown in the PERMANOVA also). It is possible to then take the principle components PC1 and PC2 and correlate them against the fatty acids to try and determine which direction the relationship is displaying in the PCA.

Additional comments

I would like to congratulate the authors on an interesting piece of work. I believe that it provides interesting knowledge on how algae respond to carbon and amino acid enrichment and on osmotrophic growth. There are a few methodological clarifications which I would like to see before publication (especially concerning the growth stage the algae were in during the long-term experiment), but I believe this paper contributes to this field. As mentioned in the review, I think this is of interest to ecologists and biotechnology researchers’ and will therefore be well received.

·

Basic reporting

Generally this is a well written manuscript, but English expression requires some attention. Some points are mentioned below but this list is not exhaustive.

line 107: stable not stabile
line 108: resuspended not dissolved
line 119: 'supernatants were' not 'was'
line 127: weighed not weighted
line 160: presumably this manuscript is derived from a thesis but this reference to a chapter should be removed
References: a number of references have genus/species names that are not italicised

Experimental design

With any work on osmotrophy it is important to ensure that the organic substrates are truly taken up by the algae themselves and rule out the possibility that bacterial breakdown of organic matter is driving the changes observed. Were the cultures axenic and how was this checked?

line 113 and line 149: It should be specified that these were independent replicate cultures, not technical replicates from a single culture (pseudoreplication)

Validity of the findings

Lines 69 -72: I am not especially convinced that buildup of triacyl glycerols (storage lipid) necessarily leads to higher levels of PUFAs. Most LC-PUFAs are found in membranes (Roessler PG (1990) Environmental control of glycerolipid metabolism in microalgae: commercial implications and future research directions. J Phycol 26:393–399) though this can change in stationary phase when LC-PUFAs can be mobilized from membranes into TAGs - see Boelen et al 2017 (Boelen, P., van Mastrigt, A., van de Bovenkamp, H.H. et al. Aquacult Int (2017) 25: 577. https://doi.org/10.1007/s10499-016-0053-6). The latter point is important in view of the long term experiment results and needs to be incorporated into the discussion

Additional comments

Overall this is an interesting study but the manuscript requires further attention before it could be considered suitable for publication.

---

## Round 0.2 · Minor Revisions

The concerns of the editor and reviewers have been very nicely addressed in the revision. There are just a few small details to be corrected before I can accept the manuscript for publication:

1) The tick and axis labels of Figure 1 and 2 are too small. If reduced for publication they will not be legible. Please increase them to sizes similar to what you have in Figure 3 and they should be fine.
2) There are still a few small issues in the text (e.g. grammar and the confusion of specific growth rate and divisions per day). I have annotated these in the text in red, with yellow highlights or with a note.

I anticipate these should be pretty easy and quick to correct.

---

## Round 0.3 · accepted · Accept

Very good. That takes care of all my concerns. The Figures look great!